# Post hoc Explanations may be Ineffective for Detecting Unknown Spurious Correlation

**Julius Adebayo**
MIT CSAIL

**Michael Muelly**
Stanford

**Hal Abelson**
MIT CSAIL

**Been Kim**
Google Research

## Abstract

We investigate whether three types of post hoc model explanations–feature attribution, concept activation, and training point ranking–are effective for detecting a model's reliance on spurious signals in the training data. Specifically, we consider the scenario where the spurious signal to be detected is unknown, at test-time, to the user of the explanation method. We design an empirical methodology that uses semi-synthetic datasets along with pre-specified spurious artifacts to obtain models that verifiably rely on these spurious training signals. We then provide a suite of metrics that assess an explanation method's reliability for spurious signal detection under various conditions. We find that the post hoc explanation methods tested are ineffective when the spurious artifact is unknown at test-time especially for non-visible artifacts like a background blur. Further, we find that feature attribution methods are susceptible to erroneously indicating dependence on spurious signals even when the model being explained does not rely on spurious artifacts. This finding casts doubt on the utility of these approaches, in the hands of a practitioner, for detecting a model's reliance on spurious signals.[1]

> *It is hard to find a needle in a haystack,*
> *it is much harder if you haven't seen a needle before (Pearl).*
> —Judea Pearl

## 1 Introduction

A challenge that precludes the deployment of modern deep neural networks (DNN) in high stakes domains is their tendency to latch onto 'spurious signals'—shortcuts—in the training data (Geirhos et al., 2020). For example, Badgeley et al. (2019) showed that an Inception-V3 model trained to detect hip fracture relied on the scanner type for its classification decision. Deep learning models easily base output predictions on object backgrounds (Xiao et al., 2020), image texture (Geirhos et al., 2018), and skin tone (Stock and Cisse, 2018).

Post hoc model explanation methods—approaches that give insight into the associations that a model has learned—are increasingly used to determine whether a model relies on spurious signals. Ribeiro et al. (2016) used LIME to show an Inception-V3 model's reliance on the snow background for identifying Wolves. Such demonstration and others (Lapuschkin et al., 2019; Rieger et al., 2020) point to post hoc explanation methods as effective tools for the spurious signal detection task. However, these results conflict with evidence that indicates that practitioners (and researchers) struggle to use explanations to identify spurious signals (Chen et al., 2021; Chu et al., 2020; Alqaraawi et al., 2020; Adebayo et al., 2020; Poursabzi-Sangdeh et al., 2018). We seek to resolve this conflict by answering the simple but important question:

*Can post hoc explanations help detect a model's reliance on **unknown** spurious training signal?*

**Motivating Example.** Consider a machine learning (ML) engineer whose job is to train DNN models to detect knee arthritis from radiographs. She—the engineer—is handed a trained ResNet-50 model, to be deployed in a hospital, that relies on a hospital tag in the radiographs to detect knee arthritis. She

---

[1]We refer readers to the longer version of this work on the arxiv. Code to replicate our findings is available at: https://github.com/adebayoj/posthocspurious

has **no prior knowledge** of the model's reliance on the spurious tags. In this work, our key concern is whether the ML engineer can use post hoc explanations to identify that the model is defective.

## 1.1 Our Contributions

We address the motivating question above in a two-pronged manner. First, we develop an actionable methodology based on the ability to carefully craft datasets to induce spurious correlation in trained models. Second, we backup this experimental design with a human subject study. Taken together, the takeaway of the work is that: *post hoc explanations can be used to identify a model's reliance on a **visible** spurious signal, provided the signal is **known** ahead of time by the practitioner.* While this conclusion may seem unsurprising, it has important implications for how post hoc explanation methods should be used effectively.

**Experimental Design & Performance Measures.** We provide an end-to-end experimental design for assessing the effectiveness of an explanation method for detecting a model's reliance on spurious training signals. We define a spurious score that helps quantify the strength of a model's dependence on a training signal. Using carefully crafted semi-synthetic datasets, we are able to train models where the ground-truth expected explanation is known. Additionally, we develop 3 performance measures: i) Known Spurious Signal Detection Measure (K-SSD), ii) Cause-for-Concern Measure (CCM), and iii) a False Alarm Measure (FAM). These measures help characterize different notions of reliability for the spurious signal detection task. We instantiate the proposed design on 3 classes of post hoc explanation types—feature attribution, concept activation, and training point ranking—where we comprehensively assess the performance of these approaches across 3 datasets (2 medical tasks, and dog species classification task), and different model architectures.

When the spurious signal is known, we find that the feature attribution methods tested, and the concept activation importance approach are able to detect visible spurious signals like a text tag and distinctive stripped patterns. However, we find these approaches less effective for non-pronounced signals like background blur. The false alarm measure further indicates that feature attribution methods are susceptible to erroneously indicating dependence on spurious signals.

The cause-for-concern measure quantifies the similarity between explanations of 'normal' inputs derived from spurious and normal models when the spurious signal is unknown. Across the settings considered, we find that the methods tested are unable to conclusively detect model reliance on unknown spurious signals.

**Blinded Study.** The findings from our empirical assessment question the reliability of the methods tested; however, it might not correlate with utility in the hands of practitioners. To address this issue, we conduct a user study where practitioners are randomly assigned to one of two groups: the first group is told explicitly of potential spurious signals, and the second is not. We consider three different kinds of explanation methods along with a control where only model predictions are shown. We find that when participants are not provided with prior knowledge of the spurious signal, none of the methods tested are effective, in the hands of the participants, for detecting model reliance on spurious signals. More surprisingly, even when the participants had prior knowledge of the spurious signal, we find evidence that only the concept activation approach, for visible spurious signals, is effective. These findings cast doubt on the reliability of current post hoc tools for spurious signal detection.

**Guidance.** On the basis our analysis, we can provide the following guidance for using the approaches tested, in this work, for detecting model reliance on spurious signals **when the signal of interest is visible**:

- **Feature Attributions**: to confirm that a model is relying on a 'visible' spurious signal, the practitioner needs to obtain attributions for inputs that contain the hypothesized spurious signal, and the attribution should be computed for the output class to which the spurious signal is aligned.

- **Concept Activation**: the spurious concept should be known ahead of time, and tested against the output class to which the concept is aligned.

- **Training Point Ranking**: an input that contains the hypothesized spurious signal of interest should be used at test-time in computing training point ranking.

## 1.2 RELATED WORK

This paper belongs to a line of work on assessing the effectiveness of post hoc explanations methods (Alqaraawi et al., 2020; Adebayo et al., 2020; Chu et al., 2020; Hooker et al., 2019; Meng et al., 2018; Poursabzi-Sangdeh et al., 2018; Tomsett et al., 2020). Here we focus on directly relevant literature, and defer an extensive discussion of the literature to the Appendix.

This work departs from previous work in two ways: 1) we focus exclusively on whether these explanations can be used by a practitioner (or researcher) to detect spurious signals that are unknown to her at test-time, and 2) we move beyond sole focus on the feature attribution setting to test concept activation and training point ranking methods.

Han et al. (2020) and Adebayo et al. (2020) find that certain kinds of feature attributions and training point ranking via influence functions are able to detect a model's reliance on spurious signals. However, in their setting, the spurious signal is known ahead of time. More recently, Zhou et al. (2021) conduct an extensive assessment of several feature attribution methods also under the spurious correlation setting, for visible and non-visible artifacts, and find that these class of methods are not effective for non-visible artifacts. In addition, they also propose an experimental methodology for controlling model dependence on training set features, which allows them to quantify attribution effectiveness precisely. Overall, our findings align with theirs; however, we focus, specifically, on the setting where the spurious signal is not known ahead of time.

Kim et al. (2021) conduct an assessment of several feature attribution methods using a synthetic evaluation framework where the ground-truth explanation is known reasoning tasks. They find that feature attribution methods often attribute irrelevant features even in simple settings, and show high variability across data modalities and tasks. Plumb et al. (2021) introduce a method to identify important associations that a model might have learned, detect which of these associations are spurious, and propose a data augmentation procedure to overcome the reliance. Nguyen et al. (2021) conduct a large-scale user study to assess the effectiveness of feature attribution methods on image tasks. They find that feature attributions are not more effective than showing end users nearest neighbor training points. In this work, we only consider image tasks, however Bastings et al. (2021) devised a similar experimental procedure and metrics to test several attribution methods for spurious signal detection in text settings. They find that the effectiveness of an attribution method depends on the task, spurious signal, and other dataset dependent properties.

## 2 EXPERIMENTAL METHODOLOGY

In this section, we setup our experimental methodology. We discuss quantitative analysis of post hoc explanations derived from models trained to rely on pre-defined spurious signals, and a blinded user study that measures the ability of users to use the post hoc explanation methods tested to detect model reliance on spurious signals. We discuss the types of spurious signals considered, define a spurious score that allows us to ascertain that a model indeed relies on a signal as basis of its classification decision, and layout performance measures that capture the reliability of the explanation methods. We conclude with an overview of the methods tested, datasets, and models.

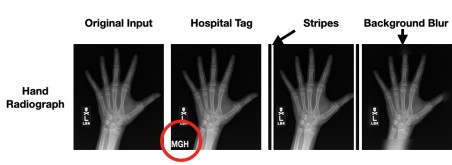

Figure 1: **Overview of Spurious Signals Considered.**

## 2.1 EXPERIMENTAL DESIGN

**Spurious Signals & Score.** We consider a spurious signal to be input features that encode for the output but have 'no meaningful connection' to the 'data generating process' (DGP) of the task. A hospital tag present in a hand radiograph is not clinically relevant to the age of the patient. If the tag encodes for the output then it is a spurious signal. Domain expertise is ultimately required for adjudicating that a signal is spurious.

We consider 3 (2 visible and 1 non-visible) kinds of spurious signals (See Figure 1): i) a localized tag; ii) a distinctive stripped pattern; and iii) Gaussian blur applied to the image background. The signals are all spatially localized, so we can easily obtain ground-truth expected explanations.

To induce reliance on spurious signals, we train models on "contaminated" versions of the training set. Given input-label pairs, $\{(x^i, y^i)\}_i^n$, where $x^i \in \mathcal{X}$ and $y^i \in \mathcal{Y}$, we can learn a classifier, $f_\theta$, via empirical risk minimization (ERM) that corresponds to minimizing a loss function, $\ell$: $\arg\min_\theta \sum_{i=1}^n \ell(x^i, y^i; \theta)$. To contaminate the training set, we apply a spurious contamination function (SCF) to the training set; $\text{SCF} : \mathcal{X} \times \mathcal{Y} \times \mathcal{C} \to \mathcal{S}$, where $\mathcal{C}$ is the spurious signal set and $\mathcal{S}$ is the transformed set. An example of an SCF is a function that pastes an hospital tag onto the bone age radiographs of all pre-puberty individuals in the dataset. To derive models reliant on a spurious signal, $c_i \in \mathcal{C}$, we simply learn a new classifier via ERM on the modified dataset as follows: $\arg\min_\theta \sum_{i=1}^n \ell\big(\text{SCF}(x^i, y^i, c_i)\big)$ to obtain $\theta_{\text{spu}}$. Contemporary evidence suggests that this approach produces models that easily latch onto the spurious signal (Nagarajan et al., 2020).

We focus on the classification setting, and restrict spurious signals to encode, only, for a single class—the *spurious aligned class*. We measure a model's reliance on the spurious signal via a score.

**Definition 2.1.** (Spurious Score). Given a spurious signal, $c_i$, the index of its spurious aligned class, $j \in [k]$, a model, $\theta_{\text{spu}} : \mathbb{R}^d \to \mathbb{R}^k$, where $\arg\max(\theta_{\text{spu}})$ indicates the classifier's predicted class, we define the spurious score as:

$$\text{SC}_{c_i, j}(\theta_{\text{spu}}) \coloneqq \mathbb{P}_{\{x^i | \theta_{\text{spu}}(x^i) \mathrel{!=} j\}}[\arg\max(\theta_{\text{spu}}(\text{SCF}(x^i, y^i, c_i))) = j].$$

Given an input that does not contain the spurious signal, and for which the model's prediction is not the spurious aligned class, the model's spurious score is the probability that the model assigns the input to the spurious aligned class if the spurious signal is added to the input.

**Model Conditions.** We focus our analysis on two model conditions: i) a 'normal model', $f_{\text{norm}}$, for which we can rule out dependence on any of the spurious signals tested across all classes on the basis of the spurious score, and ii) a 'spurious model', $f_{\text{spu}}$, for which one of the spurious signals encodes for a particular output class. We empirically estimate the spurious score and term models that have a score above $0.85$ for any of the pre-defined signals 'spurious models'. We term a model 'normal' if the spurious score is below $0.1$ across all classes and the 3 pre-defined spurious signals.

**Spurious Signal Detection Reliability Measures.** Equipped with spurious ($f_{\text{spu}}$) and normal ($f_{\text{norm}}$) models, we are now able to quantitatively assess the motivating question of this work. We do this by comparing explanations derived from spurious models, $f_{\text{spu}}$, to those derived from normal models ($f_{\text{norm}}$). We can partition the kinds of inputs used for deriving explanations into two: 1) *spurious inputs* ($x_{\text{spu}}$)—inputs that include the spurious signal and 2) *normal inputs* ($x_{\text{norm}}$)—inputs do not not contain the spurious signal. Comparing the explanations produced by these two classes of inputs for normal and spurious models, we derive reliability performance measures.

- **Known Spurious Signal Detection Measure (K-SSD)** - measures the similarity of explanations derived from *spurious models on spurious inputs* to the ground truth explanation. The ground truth explanation is one that only assigns relevance to the spurious signal as explanation of the output of a spurious model on a spurious input. K-SSD measures method reliability when the spurious signal is known. Given a similarity metric, $S_d$, then K-SSD corresponds to: $S_d\big(\mathrm{E}_{f_{\text{spu}}}(x_{\text{spu}}), x_{\text{gt}}\big)$; where $\mathrm{E}_{f_{\text{spu}}}(x_{\text{spu}})$ are explanations derived from the spurious model for spurious inputs, and $x_{\text{gt}}$ is the ground truth explanation. The similarity function, $S_d$, depends on the type of explanation considered—we will make our choice of this function concrete shortly.

- **Cause-for-Concern Measure (CCM)** - measures the similarity of explanations derived from *spurious models for normal inputs* to explanations derived from *normal models for normal inputs*: $S_d\big(\mathrm{E}_{f_{\text{spu}}}(x_{\text{norm}}), \mathrm{E}_{f_{\text{norm}}}(x_{\text{norm}})\big)$. This measure simulates the setting where a practitioner does not know the spurious signal, and can only inspect explanations for inputs without the signal. If this measure is high, then it is unlikely that such a method alert a practitioner that a spurious model exhibits defects.

- **False Alarm Measure (FAM)** - measures the similarity of explanations derived from *normal models for spurious inputs* to explanations derived from *spurious models for spurious inputs*: $S_d\big(\mathrm{E}_{f_{\text{norm}}}(x_{\text{spur}}), \mathrm{E}_{f_{\text{spu}}}(x_{\text{spu}})\big)$. We also introduce a variant of this measure, FAM-GT,

which measures the similarity of a explanations derived from *normal models for spurious inputs* to the ground truth explanation of a spurious model for that spurious input. If this measure is high, then that approach is more likely to signal to a practitioner that a model is relying on spurious signal when the model does not.

Having defined the metrics above, it remains which similarity function to use.

**Computing Metrics for Feature Attribution.** For feature attribution methods, we use the Structural Similarity Index (SSIM). SSIM measures the visual similarity between two images. Concretely, given a set of normal inputs, we obtain a corresponding spurious set of these inputs by applying the spurious contamination function, SCF to these inputs. Consequently, we can then compute the K-SSD, CCM, and FAM metrics given these two sets of inputs using the SSIM metric.

**Computing Metrics for Concept Activation.** We measure comparison between two concept rankings using a Kolmogorov-Smirnoff (KS) test comparing two distributions where the null hypothesis is that the two distributions are identical; we set significance level to be $0.05$.

**Computing Metrics for Training Point Ranking.** Recently, Hanawa et al. (2020) introduced the Identical class metric' (ICM), which is the fraction of the top training inputs, for a given test example, that belong to the same class as the true class of the test example in question. Here we also use the KS test to compare the ICM distributions for two different models and set the significance level to be $0.05$.

Taken together, these measures provide a comprehensive overview of an explanation method's performance for detecting spurious signals.

## 2.2 BLINDED STUDY

To complement the quantitative setup, we further designed a user study (IRB approved) to assess the ability of end-users (200 in total) to use post hoc explanations to detect a model's reliance on spurious signals. About 50 percent of the participants had trained a ML model before, and 74 percent had interacted with an ML model. We refer to the appendix for additional details.

**Task & Setup:** The study participants were tasked with assessing model reliability with the aid of model explanations. The participants were randomly assigned to one of two groups: the first group is told explicitly of potential spurious signals, and the second is not. They were asked to rate how likely they are to recommend the model for deployment using a 5-point Likert scale, and a rationale for their decision. Our study design follows that of Adebayo et al. (2020); however, we use a mixed within-subjects and between subjects design for the factors of interest. The Likert scale is as follows, 1: Definitely Not to 5: Definitely. Participants select 'Definitely' if they deemed the dog breed classification model reliable. We refer to Appendix I for discussion on user recruitment, statistical analyses, and study design.

**Methods:** We test SmoothGrad, TCAV (a concept activation method), a training point ranking method, and a Control setting with no explanations.

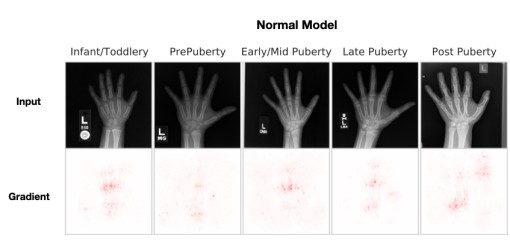

Figure 2: **Feature Attributions.** Here we show 5 different inputs, one for each bone age category, and the corresponding Gradient feature attribution map. We refer to the Appendix for an equivalent visualization for other feature attribution methods considered. We test three additional feature attribution methods: SmoothGrad, Integrated Gradients, and Guided BackProp.

## 2.3 EXPLANATION METHODS, DATA, & MODELS

Here we give an overview of the explanation methods, datasets, and models. We present a discussion of how these methods are used in practice in Appendix.

**Feature Attributions** assign a relevance score for each dimension of an input towards an output. We consider: **Input-Gradient, SmoothGrad, Integrated Gradients (IG), and Guided Backprop (GBP)**.

**Concept-Based** ((Bau et al., 2017; Kim et al., 2018)) approaches measure the dependence of a DNN's prediction on user-defined features—termed concepts. We select TCAV as the approach to assess in this class (Kim et al., 2018).

**Training Point Ranking via Influence Functions (Koh and Liang, 2017).** This approach ranks the training samples in order of importance/influence on the loss (or prediction) of a test example.

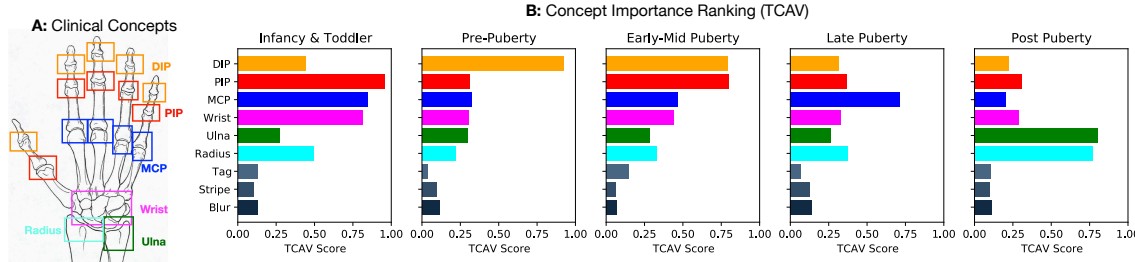

Figure 3: **Concept Importance Methods for a Normal (non-spurious) model.** Here we show the TCAV score for all clinical concepts as well as the spurious concepts for a normal model that was confirmed to not rely on the spurious signals.

**Models, Datasets, & Tasks.** We consider two medical datasets: Hand (Halabi et al., 2019) and Knee radiographs (Chen et al., 2019) and a dog breed classification task. We consider a small DNN (6 layers, 5 of which are the traditional conv-relu-batchnorm-maxpool combination) inspired by the CBR-Tiny architecture of Raghu et al. (2019) and a ResNet-50 model (See Appendix for additional details).

## 3 FEATURE ATTRIBUTIONS

In this section, we present results on whether feature attributions are effective for detecting unknown spurious correlation.

**Setup.** We consider 3 kinds of spurious signals which we term: *Tag* for the 'MGH' hospital tag added to the pre-puberty class, *Stripe* for the paired vertical stripped signal, and *Blur* for the background blur. Given these signals, we then compute the three performance measures of interest: K-SSD, CCM, FAM, and FAM-GT. K-SSD indicates a method's reliability when the spurious signal is known, CCM when the signal is not known and the practitioner only has access to inputs that don't encode the spurious signal. Lastly, FAM and FAM-GT indicates the susceptibility of a method to false positives. An oft-used heuristic based on prior work (Adebayo et al., 2020) for interpreting SSIM scores is that SSIM scores $0.2 - 0.4$ indicate weak visual similarity, $0.5 - 0.7$ indicate moderate similarity, and $> 0.75$ high similarity. This is because two random images typically have SSIM much less than $0.1$. For example, we empirically estimate the similarity of two random ($229 \times 229$) Gaussian images to be less than $0.00023$. Even for natural images, we still find the SSIM values to be below $0.005$, which substantiates the previous heuristic.

**Results.** We show performance measures for all the feature attribution methods tested for the Tag and Blur settings in Tables 1 & 11 (See Appendix). For the tag setting, the attribution methods are indeed able to attribute to the visible spurious signal and the K-SSD measure indicates this finding with mean scores typically above 0.65 for the bone age setting. Contrary to previous findings, we find that GBP outperforms other approaches for known spurious signals. Alternatively, GBP is more suceptible to false positives based on the FAM and FAM-GT score. Across all methods, we find that these methods also seem to attribute to the spurious signal (FAM-GT $> 0.4$) even when the signal is not being relied on by the model. We observe similar findings for the strip setting as well across all tasks. The CCM measure further indicates that these methods do not indicate presence of spurious signals when the signal is unknown for both the Tag and Stripe signals. This finding, however, reverses for

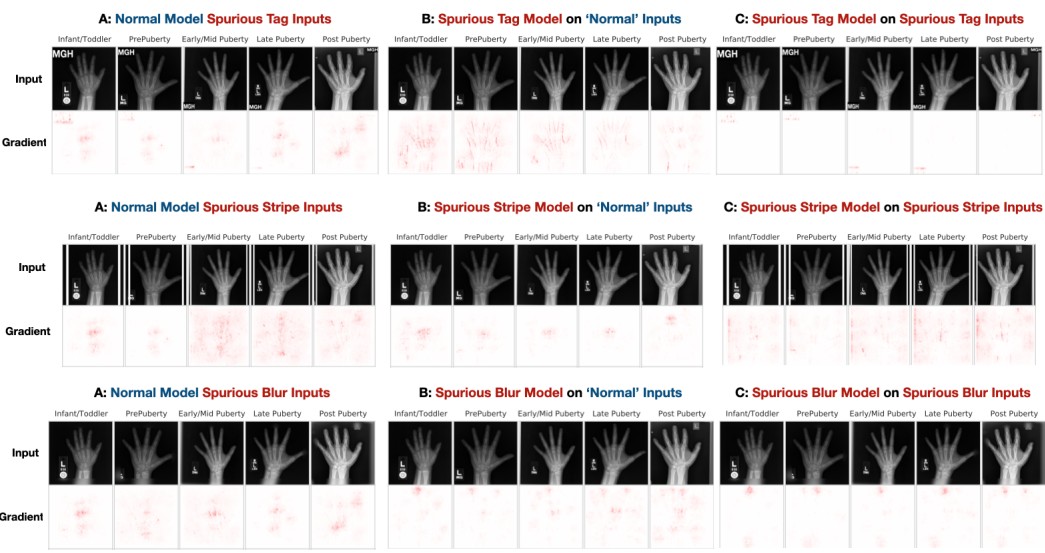

Figure 4: **Top-Detecting Spurious *Tag*.** Here we show in A) Feature attributions for 5 different inputs across the four feature attribution methods with a normal model but with spurious Tag inputs; B) Feature attributions on the same 5 inputs as in (A), but **without** spurious Tag inputs with a model that has learned a spurious alignment between Pre-Puberty and Tag; C) Feature attributions on the same 5 inputs as in (A), but **with** the spurious Tag inputs with a model that has learned a spurious alignment between Pre-Puberty and Tag. **Middle-Detecting Spurious *Stripe*.** Here we show in A) Feature attributions for 5 different inputs across the four feature attribution methods with a normal model but with spurious Stripes inputs; B) Feature attributions on the same 5 inputs as in (A), but **without** the spurious Stripe with a model that has learned a spurious alignment between Pre-Puberty and Stripe; C) Feature attributions on the same 5 inputs as in (A), but **with** the spurious Stripe with a model that has learned a spurious alignment between Pre-Puberty and Stripe. **Middle-Detecting Spurious *Blur*.** The blur images are analogous to the Tag and Stripe settings. Please refer to the Figures 7, 8, 9 in the Appendix for SmoothGrad, Integrated Gradients, and Guided BackProp examples.

Table 1: Performance metrics for each attribution method across tasks for the Tag Setting. Below each metric in the Table is another row (SEM) that indicates the standard error of the mean for each value.

| Method | Bone Age | | | | Knee | | | | Dog Breeds | | | |
|---|---|---|---|---|---|---|---|---|---|---|---|---|
| | Grad | SG | IG | GBP | Grad | SG | IG | GBP | Grad | SG | IG | GBP |
| K-SSD | 0.65 | 0.66 | 0.67 | 0.81 | 0.51 | 0.49 | 0.47 | 0.76 | 0.71 | 0.76 | 0.79 | 0.88 |
| K-SSD (SEM) | 0.0097 | 0.013 | 0.019 | 0.006 | 0.012 | 0.017 | 0.019 | 0.023 | 0.01 | 0.011 | 0.014 | 0.01 |
| CCM | 0.37 | 0.39 | 0.35 | 0.75 | 0.32 | 0.33 | 0.35 | 0.66 | 0.42 | 0.41 | 0.39 | 0.64 |
| CCM (SEM) | 0.0031 | 0.002 | 0.015 | 0.029 | 0.027 | 0.023 | 0.029 | 0.014 | 0.013 | 0.016 | 0.012 | 0.015 |
| FAM | 0.51 | 0.55 | 0.53 | 0.68 | 0.46 | 0.47 | 0.45 | 0.69 | 0.59 | 0.64 | 0.68 | 0.73 |
| FAM (SEM) | 0.0029 | 0.0019 | 0.018 | 0.024 | 0.023 | 0.024 | 0.019 | 0.016 | 0.015 | 0.011 | 0.022 | 0.035 |
| FAM-GT | 0.56 | 0.53 | 0.46 | 0.61 | 0.42 | 0.48 | 0.41 | 0.63 | 0.76 | 0.73 | 0.77 | 0.81 |
| FAM-GT (SEM) | 0.017 | 0.035 | 0.0253 | 0.028 | 0.016 | 0.019 | 0.0045 | 0.006 | 0.011 | 0.033 | 0.024 | 0.0053 |

the non-visible blur spurious signal. Across all measures, we find that all methods struggle to reliably indicate that spurious models are reliant on the blur signal.

Additionally, we also find that the FAM scores are typically higher than the CCM scores across the tasks. This finding indicates that the feature attribution methods tested are more susceptible to false positives than they are to indicate to a practitioner that a model is defective in the absence of the spurious signal, a finding that casts doubt on the utility of such approaches in practice.

## 4    CONCEPT ACTIVATION IMPORTANCE

We find that concept methods can indicate a model's reliance on Tag and Stripe signals when known. However, the approach struggles to detect Blur signal even when known. As is the case with feature attributions, when a spurious signal is not explicitly tested for, our significance tests indicate that reliance cannot be detected in the non-spurious concepts available.

**Overview & Setup.** In this setting, we compute the 3 performance metrics of interest: K-SSD, CCM, and FAM. To measure comparison between two concept rankings, we use a Kolmogorov-Smirnoff (KS) test comparing two distributions where the null hypothesis is that the two distributions are identical; we set significance level to be 0.05.

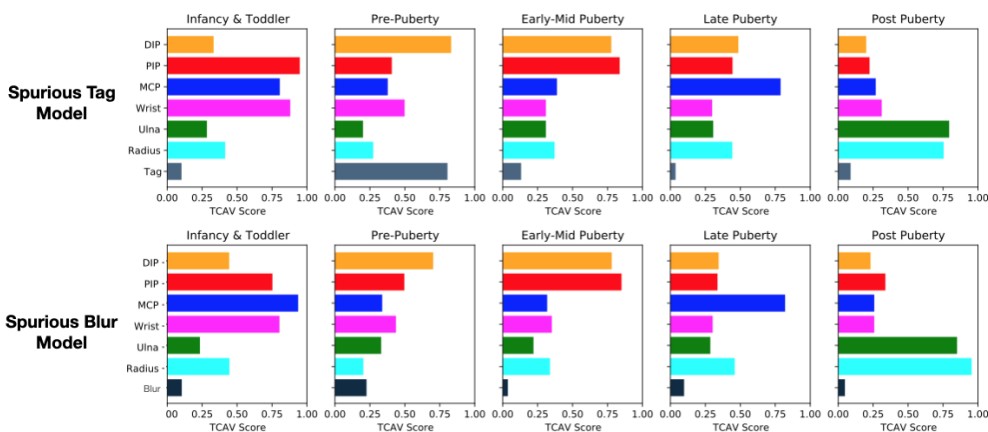

Figure 5: **Concept Results for Tag and Blur Models.** TCAV scores for a model reliant on the spurious Tag and a model reliant on Blur.

**Result.** In Figure 5, we show TCAV scores for each bone age class for both the spurious tag and the blur models. In Table 2, we show results of the KS-Test across metrics for the Tag Setting. Here, a X means we reject the null, while an ✓ means we are unable to do so at the pre-specified significance level. For the K-SSD metric, the KS-test rejects the null for the Tag and Stripe signals. However, we find that the opposite is the case for the blur signal. This suggests that concept rankings can help detect reliance on the visible signals but not non-visible signals when the spurious signals are unknown. However, for the CCM metric, we are unable to reject the hypothesis that the distribution of concept rankings are not similar for all spurious signals. This finding suggests that when the spurious signal is unknown, and we only compare the distributions of known (non-spurious) concepts for a normal model and a spurious model, there is high similarity. Lastly, a difference in means test as well as a KS-Test for the FAM measure indicates that the normal models do not rely on the spurious signals as well. Overall, this finding suggests that TCAV is less susceptible to false positives.

Table 2: Concept Metrics Tag.

| Metric | Result |
|--------|--------|
| K-SSD | X |
| CCM | ✓ |
| FAM | ✓ |

## 5    TRAINING POINT RANKING & BLINDED STUDY

**Overview & Setup.** We now describe our empirical findings for the training point ranking via influence functions approach. Here we present results for the case where the spurious signal is aligned with the Pre-Puberty class.

**Results.** The main take away is that given a known spurious signal, the fraction of top ranked training spurious signal inputs increases with a spurious model across all of the spurious signals. While this might seem encouraging, we note that such increase actually indicates that the ICM metric might provide illusory confidence in a spurious model. Ultimately, a critical requirement here is knowing what the spurious signal ahead of time and to be able to select the right inputs to inspect.

**Blinded Study.** We now turn to a summary of the results of the blinded user study. The median recommendations selected by participants (200 in total) is reported for each explanation-model condition in Table 6. We plot a box plot of all 16 categories in the appendix. Our setting mimics traditional randomized experimental settings, so we adopt a randomization inference analysis to determine the effect of each treatment, which is the explanation method in this case, on the ability of the users to recommend a model. A higher Likert score means the user is more likely to recommend a model. Consequently, if an explanation method is effective, then it should be the case that users should be less likely to recommend a model that relies on a spurious signal.

We conduct two kinds of statistical analyses of the data. First, under each condition, we perform a difference of means test for each treatment compared to the control setting. Secondly, for each model manipulation, we performed a one-way Anova test, and a Tukey-Kramer test to assess the effect of the explanation type on the ability of the participants to reject a defective model.

We observe that when blinded, in none of the methods do participants conclusively reject spurious models. Perhaps more surprising, when the participants were not blinded, we see that only participants using the TCAV approach rejected a spurious model. This finding has significant implications on whether these current tools are effective in the hands of a practitioner.

Table 3: Training Point Ranking

| Metric | Result |
|--------|--------|
| K-SSD | X |
| CCM | ✓ |
| FAM | ✓ |

| Method | B-Normal | NB-Normal | B-Spurious | NB-Spurious |
|--------|----------|-----------|------------|-------------|
| SmoothGrad | 4* | 4* | 3* | 3 |
| TCAV | 4* | 3 | 3* | 2* |
| Influence | 3* | 3 | 3* | 3 |
| Control | 4 | 3 | 4 | 4 |

Table 4: **Blinded User Study**. Here we report the median recommendation for each condition across all explanation types and control. This median is derived from user provided responses assessing model reliability. B indicates Blinded, and NB indicates Not-Blinded. In the Table, the * symbol indicates statistically significant conditions. For the sake of space, we defer the full distribution description to the Appendix.

## 6 DISCUSSION & CONCLUSION

**Conclusion.** DNNs trained on image datasets can naturally rely on spurious training signals (Geirhos et al., 2020). Discovering this reliance is crucial in consequential settings like medical imaging. Post hoc explanations methods are a promising direction towards detecting such reliance; however, their effectiveness is currently under question. We investigated whether 3 classes of post hoc explanations are effective for detecting a model's reliance on spurious training signals. We present an experimental setup that can also be easily adapted to other settings to assess a larger class of approaches. The setup comes equipped with a spurious score and performance measures for spurious signal detection. We find that the 3 classes of post hoc explanations tested are only sometimes able to diagnose the spurious training signal even if they are used to explicitly test for model dependence on these signals. Consequently, our findings calls for, potentially, a completely different paradigm of methods that are designed to address the important and challenging question of detecting spurious training signals.

**Limitations.** In this work, we have focused exclusively on DNNs trained on image tasks; related work (Bastings et al. (2021) has considered the text setting), however, it is unclear if our findings will generalize to other modalities like time-series data. While we considered a diverse set of methods, the literature on post hoc explanations is quite vast, so undoubtedly there exist methods that do not fit neatly into the 3 explanation classes that we explored.

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
