# OpenReview forum: "Post hoc Explanations may be Ineffective for Detecting Unknown Spurious Correlation"
_ICLR.cc/2022/Conference — ICLR 2022 Poster_

### Official Review · Reviewer_uWNE · 2021-10-29

**Correctness:** 3
**Technical Novelty And Significance:** 2
**Empirical Novelty And Significance:** 3
**Recommendation:** 6
**Confidence:** 3

**Main Review:**

The paper offers an analysis on three metrics for explaining post-hoc reliance of models on spurious signals. Even though at first inspection the metrics proposed seem reasonable, at the authors admission, they turn out not to be all that good detecting reliance on spurious signals, especially when the signal is unknown a priori ( admittedly the hardest case).

Issues of the paper :
- Clarity:  from the main body of the paper the analysis of the metrics is lacking, one has to refer to the appendix to get a better idea of the metrics proposed and how they are used in practice.
- Even though the authors claim a test on dog breeds dataset these are not included -- also all dogs belong to the same species Canis familiaris, the classification task refers to dog breeds , not species.
- Contributions: the paper proposed 3 metrics that as admitted by the authors are not performing as well as one would like. That on itself is not an issue as semi-negative results have merit, however, there is no indication of proposals of the authors on what the better metrics should be.



**Summary Of The Paper:**

The authors present an analysis on post-hoc explanation metrics that measure the reliance of a model to spurious signals. The paper offers insights on three metrics  K-SSD, CCM and FAM by deploying them in the analysis of DNNs  trained on medical image datasets. The authors also conduct a blinded study.

**Summary Of The Review:**

Overall this paper is an analysis paper of 3 self-proposed metrics on posthoc analysis of spurious signals. The metrics turn out not to be accurate enough for the tasks at hand, especially when the spurious signal is implicit or unknown. There is no discussion how to make the metrics better at detecting spurious signals.



###### POST REBUTTAL ####
Update the score to 6

---

### Official Review · Reviewer_Mwip · 2021-10-30

**Correctness:** 3
**Technical Novelty And Significance:** 2
**Empirical Novelty And Significance:** 3
**Recommendation:** 6
**Confidence:** 4

**Main Review:**

First of all I would like to thank the authors for this interesting piece of work. Below, I will briefly list the main strength and weaknesses of this submission and make suggestions for improvement. I hope you will find them helpful and constructive.

Strengths
- This work raises very important and timely concerns with regard to using NNs for critical application domains like health care.
- The scope of this work is very extensive: 2 synthetic and 1 empirical dataset, 3 different spurious signals, 3 (considering the different variants of feature attribution algorithms 6) different post-hoc methods

Weaknesses
- The spurious signal detection reliability measures are not explained sufficiently. How do you quantify similarity for the three spurious signal detection reliability measures? Equations would be helpful. How is the ground truth defined for the K-SSD measure, is this normal input and normal model?
- How should these scores be judged, what value do you interpret as problematic and why? Is there a rule of thumb that you can recommend?

- You should mention in the limitations (not in a food note), that the 'normal model' could still contain unknown spurious signals in the original data set

- The empirical study needs a more detailed description (maybe in the supplement). What population was recruited for the study, is this a population of medical practitioners, machine learning engineers, or random subjects? This should also be considered in the limitations. Is this representative of the people that will use these methods? Also is there reason for concern when choosing a within-subject design here. Briefly, when participants detect a spurious signal with one method, will this not influence their judgement based on the next method. Was the order of methods counter-balanced?

- The results of the empirical study should be reported properly (measures of variance across participants). Are differences between normal and spurious and blinded and not blinded statistically significant using a formal test. Also why are the results mentioned in the discussion and not the results.

- It is not clear to me, why the spurious MHG hospital tag signal appears to be present in all classes and not just the spurious (pre-puberty) class in Figure 5. Can you clarify?

- I commend the authors for doing a formal statistical test (Kolmogorov-Smirnoff) for the concept activation method. I would suggest including a visualization of these test results, maybe a bar plot of only the spurious concept and indicating which comparisons differed significantly with asterisk or lines. Of course the same would be useful for other comparisons that you report in tables or figures for example on the three performance measures across different methods that you defined. A formal test would substantiate your claims.

- To improve clarity of the figures, I would suggest adding arrows to Figure 1 to direct the readers attention to the spurious signals.

- Overall, the manuscript seems to be quite packed. I would suggest to move some results to the supplement, for example 3 of the 4 methods for feature attribution and Figure 2 may be good candidates.

-----------------------------------
Minor points

There are some typos in the manuscript. Please, have it proof-read.
- p6: “We results report for the small DNN model in the paper” => “We report results”
- p3: “which of these associations are be spurious” => are spurious



**Summary Of The Paper:**

The authors present work that aims to test, whether post hoc explanation methods can detect unknown spurious signals. They perform an analysis on two different medical image datasets to which they introduce two different types of synthetic, spurious signals: pronounced spurious signals (stripes and hospital tags) and non-pronounced signals (blurs). When testing the ability of different post hoc explanation methods to identify spurious signals, they find that feature attribution and concept activation are able to identify pronounced, but not non-pronounced spurious signals that are known. When faced with unknown signals, none of the methods that were tested appears to be able to identify them. They also provide results from an empirical study suggesting that none of the methods allowed practitioners to identify unknown spurious signals and only concept activation appears to enable practitioners to identify known spurious signals.

The main contribution of this work are the empirical results and alerting researchers (and potentially practitioners) to very substantial problems with current post-hoc explanation. The technical contributions are negligible.


**Summary Of The Review:**

Overall, this paper raises important questions about the usefulness of current post-hoc explanation approaches for NNs and provides a range of empirical results to support these claims. While the contribution is incremental compared to previous studies, I believe the authors do raise some important points that will help to stimulate more research in improving post-hoc explanation methods. However, the manuscript is quite dense, lacks some important clarifications (e.g., about the performance measures and the empirical study) and more rigorous statistical tests to substantiate the authors claims. For this reason, I cannot support acceptance of the manuscript as is, but I am willing to reconsider, should my concerns be addressed.

---

### Official Review · Reviewer_Ng3n · 2021-10-31

**Correctness:** 3
**Technical Novelty And Significance:** 3
**Empirical Novelty And Significance:** 3
**Recommendation:** 6
**Confidence:** 4

**Main Review:**

Strengths:
+ The authors develop 3 kinds of spurious signal detection reliability measures: Known Spurious Signal Detection Measure (K-SSD), Cause-for-Concern Measure (CCM), False Alarm Measure (FAM).
+ The authors carefully design the experiments to assess the effectiveness of an explanation method for detecting a model’s reliance on spurious training signals.
+ The authors further conduct a user study to measure the ability of users to use the post hoc explanation methods tested to detect model reliance on spurious signals.
+ The authors' findings indicate that the explanation methods are less effective to detect non-visible spurious signals and only the concept activation approach for visible spurious signals is effective, which cast doubt on the reliability of current post hoc tools for spurious signal detection.

Weaknesses:
- The paper is somewhat hard to follow.
-- The authors claim that they find that these methods also seem to attribute to the spurious signal (FAM > 0.5) even when the signal is not being relied on by the model. It is unclear whether "the model" here is the normal models or spurious models. FAM > 0.5 means that both spurious model and normal model generate similar explanations on spurious inputs, but it is unclear whether the similarity of the explanation comes from the spurious signals or not, so we don't know whether the normal model attributes the spurious signals.
-- The authors claim that (Section 3) As indicated, the CCM measure further indicates that these methods do not indicate the presence of spurious signals when the signal is unknown for both the Tag and Stripe signals. It is unclear how the authors get this conclusion from low CCM values.  We only know that (Section 2.1) If this measure (CCM) is high, then it is unlikely that such a method alert a practitioner that a spurious model exhibits defects. In Table 1, GBP has high CCM while other methods have low CCM. Why do the authors think that the CCM measure indicates that all these methods do not indicate the presence of spurious signals even with both high and low CCM?
-- The authors did not report the K-SSD, CCM, FAM, and KS-test values in Section 4, which makes some discussions hard to follow. For example, the last sentence in Section 4: "Lastly, a difference in means test for the FAM measure indicates that the normal models do not rely on the spurious signals. Overall, this finding suggests that TCAV is less susceptible to false positives".

- Tables 1, 2 display the K-SSD, CCM, FAM values in different settings. These values can describe the difference of explanations in various settings (e.g., normal and spurious models). However, even for the same model trained on the same datasets with different initial states, the outputs and explanation results might be different. It would be better if the authors displayed the values of K-SSD, CCM, FAM of explanation among the same models trained with different initial states on the same inputs. With the results as a comparison, it would be easier for readers to understand how much the spurious model or spurious inputs affect the explanation results.

- It would be better if the authors displayed the standard variance of the measurements (e.g., Table 1, 2).

- Minor: In Table 1,2, FAM values are always much higher than CCM, which means spurious model and normal model generate more similar explanations on spurious inputs than normal inputs. It is worth some discussion.

- Minor: GBP achieve much higher CCM and FAM scores than other methods (Table 1, 2). It would be better if the authors gave more discussion about it.

**Summary Of The Paper:**

The paper aims to validate whether post hoc explanation methods are effective for detecting unknown spurious correlations. The authors design 3 kinds of spurious signal detection reliability measures: Known Spurious Signal Detection Measure (K-SSD), Cause-for-Concern Measure (CCM), False Alarm Measure (FAM). Based on the 3 measurements, the authors conduct extensive experiments to validate the reliability of 3 kinds of post hoc explanation methods for detecting spurious correlations.

**Summary Of The Review:**

The overall idea of the paper is good. But it is hard to follow and some analyses are not clear.

---

### Public Comment · ~Tylor_James1 · 2022-09-04
**hey**

This investigation shows the importance of spurious correlation and you can find attribution methods from open review. The https://www.essayontime.co.uk/case-study/ is useful for readers to learn more from these reviews. Check for more information related to this.

---

> ### Public Comment · ~Raymond_Li2 · 2022-10-04
> **Problem with Link**
>
> The link is not working ...

---

### Decision · Program_Chairs · 2022-01-20

**Decision:**

Accept (Poster)

**Comment:**

This paper demonstrates that current post-hoc methods to explain black-box models are not robust to spurious signals based on three metrics especially when the spurious signals are implicit or unknown.  Technical novelty is limited because the paper presents primarily empirical results instead of novel machine learning techniques. However, the problem is very important and timely, and significance to the field and potential impact of the presented results to advance the field are high as reviewers emphasized. There are ways to further improve the paper, including the clarity of presentation, although the authors improved in the revised manuscript.  Overall, this paper deserves borderline acceptance.